# Rapid Methadone Metabolism in Opioid Use Disorder: A Case Report of Clinical Challenges and Individualized Treatment Approaches

**DOI:** 10.3390/reports8040262

**Published:** 2025-12-12

**Authors:** Farhana Nazmin, Jaskaran Singh, Narges Joshaghani, Elisio Go

**Affiliations:** 1Psychiatry, Bronxcare Health System, Bronx, New York, NY 10456, USA; njoshagh@bronxcare.org (N.J.); eliseogo@bronxcare.org (E.G.); 2Psychiatry and Behavioural Sciences, Nassau University Medical Center, East Meadow, New York, NY 10456, USA; jaskaran.4997@gmail.com

**Keywords:** rapid methadone metabolism, methadone treatment, clinical challenges, individualized treatment approaches, case report, individualized treatment approaches

## Abstract

Rapid methadone metabolism in patients with opioid use disorder could complicate methadone treatment. Toxicology screenings to monitor methadone levels may show negative for methadone, even with regular adherence to a regimen. A patient receiving treatment for opioid use disorder tested negative for methadone in 11 out of 22 toxicology screenings (50.0%). We hypothesized that the patient was a rapid methadone metabolizer. After tapering doses to a maintenance level and using supervised urine collection, the patient was negative for methadone in seven out of seven tests (100.0%), but positive for cocaine in five out of seven tests (71.4%) near the end of the maintenance period. Chronic cocaine use and genetic factors, particularly CYP2B6 polymorphisms, have been found to cause rapid methadone metabolism. Clinicians should be vigilant for unusual metabolic reactions and modify dose and monitoring schedules accordingly. More investigation into the physiological and genetic aspects of methadone metabolism is needed.

## 1. Introduction

Opioid use disorder is a problem worldwide. Every year, opioid-related deaths increase. There are different treatment options for opioid use disorder. Like heroin or opium, methadone is an opioid. Since the 1950s, methadone maintenance therapy has been used to treat opioid dependence [1]. The patient who is dependent on opioids takes a pill or liquid form of methadone once a day, which lessens their opioid cravings and withdrawal symptoms [2]. Intensive outpatient programs (IOPs) for substance abuse are direct services for individuals with co-occurring mental and substance use disorders or substance use disorders without a history of medical detoxification or round-the-clock supervision. IOPs are substitutes for residential and inpatient care [3]. Patients regularly follow up at these clinics to continue their regular lifestyle to be minimally affected by opioid withdrawal. The clinics are intended to provide coping mechanisms, relapse prevention, and psychosocial support. The number of times a person needs to pick up doses from a methadone clinic is reduced to three times per week after the initial 90-day period. Every time a participant comes in for treatment, they are given a two-day supply of methadone because it is a daily requirement. If the patient maintains their regularity, the visits can be further decreased to once a week, followed by twice a month, and lastly once a month, but it takes a long time to reach that level. CYP3A4 and CYP2B6 are the major CYP isoforms involved in methadone metabolism, but CYP2D6 has minimal impact on methadone dosage requirements [4]. We present a unique case of a patient exhibiting rapid methadone metabolism without experiencing opioid abstinence syndrome (OAS) upon presentation at a methadone clinic. This case discusses the importance of individual treatment plans and the challenges in treating patients with typical metabolic responses to methadone. This will add to clinical practice knowledge by informing dosing strategies and monitoring protocols, ultimately improving patient outcomes in Opioid Use Disorder (OUD).

## 2. Methadone Metabolism and Serum Concentration

Methadone metabolism, monitoring serum levels, and peak concentrations provide critical insights into the pharmacokinetic profile of the patient. Individual differences in methadone pharmacokinetics are quite significant. Methadone’s half-life varies from 8 to 59 h based on the patient. The average time is 24 h [5]. Thus, methadone induction should start low and increase gradually over a few days or weeks while being monitored every day. Serum levels at steady daily doses peak two to four hours after dosing and then gradually drop, allowing for a full day without overdosing or withdrawal [6]. CYP2B6, CYP2D6 and the liver’s CYP450 3A4 enzymes are primarily involved in metabolizing methadone [7]. Draw peak and trough blood specimens approximately 3 h and 24 h after dose administration, respectively, to measure serum methadone levels. Serum methadone levels and dose generally correlate, but because patient response varies greatly, there is no established therapeutic window based on serum methadone levels [8]. Methadone minimum trough levels between 300 and 400 ng/mL may be linked to a lower risk of heroin use; however, the patient’s overall response should be the primary consideration when determining the appropriate therapeutic dose, not the patient’s serum plasma levels. Peak/trough ratios greater than 2:1 might point to a quick metabolism [9].

## 3. Case Discussion

The patient is a 44-year-old Hispanic female with a psychiatric history notable for substance use disorder, including opioid, cocaine, and tobacco use disorder, and substance-induced mood disorder. Her treatment history includes one prior detoxification and rehabilitation attempt, but no long-term inpatient residential programs or outpatient addiction services. She has no history of prior suicide attempt/self-injurious behavior. However, she has had three unintentional drug overdoses in the past while seeking to get high, leading to admission into the hospital. Her medical history is significant for rheumatoid arthritis, gastric ulcer, celiac disease, and migraine headaches. She presented to the opioid treatment program (OTP) clinic for methadone maintenance therapy to address her opioid use disorder.

## 4. Substance Use History

The patient began smoking cigarettes during her teenage years, initially starting with a few cigarettes and then gradually increasing to one pack per day. She currently smokes approximately two or three cigarettes daily. At the age of 36, she began abusing Tylenol # 3 (Tylenol-Codeine) for arthritis pain, but no longer engages in this behavior. She was prescribed oxycodone pills for a toothache at the age of 38, which marked the onset of her opioid addiction. She reported abusing Percocet, originally prescribed by her dentist, leading to a 9-month history of oxycodone misuse. During this period, she consumed approximately 9–10 mg of oxycodone pills orally. The patient began sniffing cocaine around the same time she started purchasing oxycodone from the street. She also has a history of intermittent benzodiazepine use, as evidenced by positive toxicology results. She illustrated that the last use of oxycodone was 6 years ago, when she was discharged from the hospital after an unintentional overdose on oxycodone pills to alleviate withdrawal symptoms. The patient denied any history of heroin use, recreational substance use, or intravenous drug use. She reports maintaining sobriety from opioids by consuming methadone daily and actively participating in group sessions and meetings at OTP.

## 5. Psychiatric History

The patient was first admitted to a psychiatric hospital for depression and the ingestion of 8 to 10 mg of Oxycodone Pills 6 years ago. She reported that she was self-medicating but denied any suicidal intent. Additionally, she disclosed unintentional overdoses with Percocet twice and once with methadone in the past, which led to admission into the ED (Emergency Department) or inpatient medical/psychiatric facilities. She was diagnosed with substance-induced mood disorder and prescribed zoloft, trazodone, and melatonin in the past, but she is not currently taking any psychotropic medications. She has never experienced anxiety, mood swings, or any perceptual disturbances.

## 6. Medical and Surgical History

The patient has a medical history of migraine headaches, rheumatoid arthritis, gastric and duodenal ulcers with duodenal stricture, and celiac disease. Her surgical history includes a Cholecystectomy and Tubal ligation. She used to take Topamax tablets and amitriptyline for migraine headaches, prescribed by her Neurologist, and Gabapentin for hand joint pain. However, she is not currently taking any medication.

## 7. History of Presenting Illness and Clinical Course Along with Lab Findings

The patient has been sober since her discharge from the psychiatric hospital 6 years ago, when she was referred to the OTP Clinic to initiate methadone maintenance treatment. Based on examination and assessment at the OTP, it was determined that the patient had OUD for at least one year and was diagnosed with opioid dependence. She was started on a six-day-a-week schedule (Monday to Saturday clinical visits and take-home bottle for Sunday). She began methadone induction at a low dose (10 mg daily) with titration accordingly. EKG findings on admission date revealed a normal sinus rhythm and QTc 428 milliseconds.

During her treatment, the patient reported experiencing opiate cravings and withdrawal symptoms in the early morning, including sweating, body aches, nausea, discomfort, and persistent cravings for opioids. During subsequent visits, her methadone dosage was gradually increased from 10 mg to 100 mg, at which point she no longer reported any cravings or withdrawal symptoms. Toxicology screenings conducted for 16 months after admission showed negative results for methadone in 11 out of 22 tests. Eleven tests were positive for methadone, and one test was positive for cocaine with a low urine creatinine level of 11 mg/dL.

The patient was asked about negative methadone in her urine toxicology, and she mentioned that she takes methadone regularly without any skips in the schedule. Even on 12 April 2019, the patient was on supervised urine submission and still came out negative for methadone consistently. So we ordered serum methadone level and urine toxicology for comparison on 17 April 2019 to determine the presence of methadone or not and rule out tampering with urine toxicology. The results showed negative methadone in urine toxicology, and no methadone was detected in the blood. The patient’s urine collection was supervised on 23 April 2019 and still came out negative for methadone and opiates. The patient seemed to be a metabolizer of methadone, hence serum methadone levels were repeated again at peak level 2–3 h after dosing, which showed no detected trough level on 17 April 2019 and a peak level of 170 on 2 May 2019. ECG (9 May 2019): NSR @ 77, QTc 448.

In June 2022, the patient stated she was ready to start being tapered down with eventual discontinuation of methadone and would like to start with a 10 mg decrease from a 100 mg dose gradually. She also stated her intention to continue to maintain sobriety regarding heroin and other illicit drugs. Currently, the patient feels well; she denies opiate withdrawal symptoms and depression, and ison 5 mg per day without complaint of withdrawal symptoms or craving/relapse. Urine toxicology from 2 January 2024 to 14 March 2024 was positive for cocaine five out of seven times and all negative for methadone and other opiates. She has two low urine creatinine levels of 4 and 18, respectively, on two occasions, as explained in Table 1. The patient appeared to be a fast metabolizer of her methadone based on previous assessments with serum methadone levels. Moreover, the following substances, such as benzodiazepine and cannabis, were detected occasionally, while methadone showed intermittent detection. During the maintenance therapy in early 2024, cocaine was observed to be positive, followed by a low creatinine level. Mostly, tests were negative for these substances (Table 1).

## 8. Important Lab Findings

Urine toxicology across treatment phases revealed persistently negative methadone screens despite supervised dosing, indicating rapid metabolism. Occasional cannabis and intermittent cocaine positivity were observed, with no consistent opiate or benzodiazepine detection. Low urine creatinine suggested sample dilution during the maintenance phase. Cumulatively, the findings indicate stimulant use during the tapering periods and rapid methadone clearance. All detail is summarized in Table 2.

## 9. Discussion

This is a unique case as the patient, even being a rapid metabolizer, was stable on a relatively usual dose of methadone. During the induction phase methadone overdose, the patient may have experienced transient metabolic inhibition causing the overdose associated toxicity. Fifty percent of the urine toxicology tests have negative results. The low urine creatinine levels indicate that the samples have been diluted. The rapid methadone metabolism is confirmed in the serum results indicating that there is no non-adherence or tampering. A similar case was found by Hobbins et al., but that patient, being an ultra-rapid metabolizer, required a daily methadone dose of 1200 mg, split 400/400/400 [10], and showed symptoms of opioid abstinence syndrome (OAS) when a typical dose was given, which is not seen in our patient. Our patient can have multiple factors that can cause the rapid metabolism without symptoms of OAS. One of the factors can be genetic allelic variation. The primary CYP2B6 metabolizer of methadone is an individual’s phenotype, which can impact serum methadone levels [[11],]. Homozygotes of the highly active CYP2B6*5 allele will metabolize methadone quickly and with little chance of toxicity [12]. So, this can be a case of CYP2B6*5 polymorphism. Another cause may be regular use of cocaine, which may affect the levels of methadone. Regular cocaine users demonstrated a trend toward a lower AUC (*p* = 0.09) and faster methadone clearance (*p* = 0.08), as well as a significant decrease in C (min) (*p* = 0.04). Because it reduces methadone exposure, regular cocaine use may hurt treatment outcomes for OUD in people receiving methadone maintenance [13]. Demographic causes, such as age, sex, and ethnicity, may be thought of as a few other factors contributing to rapid metabolism. The metabolism of (R)-methadone and (S)-methadone was significantly impacted by female sex (*p* = 0.016 and *p* = 0.044, respectively) [14]. The metabolism of (S)- methadone was significantly impacted by CYP2B6 loss-of-function (LOF) alleles (*p* = 0.012) [14]. The metabolism of (R)-methadone was significantly impacted by body mass index (BMI) (*p* = 0.034). Males, those with LOF alleles, and those with higher body mass index seemed to have lower methadone metabolism [14]. Since the case centered on the investigation of clinical correlation and pharmacokinetic response, we did not conduct a genetic analysis. Considering the patient’s stable condition on a standard methadone dose, the potential genetic, behavioral, and demographic factors discussed—such as the presence of the CYP2B6*5 allele, regular cocaine use, and individual demographic characteristics—warrant further investigation.

## 10. Limitations

The absence of the genetic analysis component of the case is explained by the limited available case-specific pharmacogenetic resources, as well as the case’s clinical focus. This single-patient report is limited by the absence of genotyping, as well as the possible confounding variable of concurrent cocaine use, which may have affected the metabolism of methadone.

## 11. Conclusions

The intricacy of methadone metabolism and its consequences for treating opioid use disorder are highlighted in this case. The patient’s quick methadone metabolism, combined with the absence of the anticipated opioid abstinence syndrome, emphasizes the need for individualized treatment plans in methadone maintenance therapy. Clinicians need to be on the lookout for unusual metabolic reactions and modify their dose and monitoring schedules accordingly. More investigation into the physiological and genetic aspects affecting methadone metabolism may yield deeper understandings and, ultimately, more efficient and customized treatments for those suffering from opioid use disorder. This instance adds to the increasing amount of data demonstrating the necessity for a comprehensive comprehension of methadone pharmacokinetics in a range of patient demographics. This case highlights the significance of individualized monitoring and titration of potentially problematic methadone doses. The integration of available pharmacogenetic testing can identify high-risk individuals and help design personalized methadone maintenance approaches that maximize safety and efficacy.

## Figures and Tables

**Table 1 reports-08-00262-t001:** Results of the patient’s urine toxicology screening tests during treatment.

Urine Testing Period ^a^		Methadone	Benzodiazepine	Cannabis	Opiates	Cocaine	Notes
Months 1–6	July to December 2019	(+) 2× (−) 2×	(+) 4×	(+) 4×	(+) 2×	(−)	Negative for other illicit substances
Months 6–12	January to June	(+) 1× (−) 2×	(+) 2×	(+) 2×	(+) 1×	(−)	Negative for other illicit substances
Months 12–18	July to December 2020	(+)	(−)	(+) 1×	(−)	(−)	Negative for opiates and other illicit substances
Months 18–24	January to June 2021	(−) 7×	(−)	(−)	(+) 1×	(−)	Negative for methadone in 7 tests
Months 24–30	July to December 2021	(−)	(−)	(−)	(−)	(−)	Negative for methadone in 5 tests
Months 30–36	January to June 2022	(+) 2×	(−)	(−)	(−)	(−)	Positive for methadone in 2 tests
Months 36–42	July to December 2022	(−)	(−)	(−)	(−)	(−)	Negative for methadone in 6 tests
Months 30–36	January to December 2023	(−)	(−)	(−)	(−)	(−)	Positive for methadone in 2 tests
Maintenance period	January to June 2024	(−)	(−)	(−)	(−)	(+)	Low urine creatinine (4, 18) in 2 tests Positive for cocaine in 5 of 7 tests

Results of the Patient’s urine toxicology screening tests during treatment. ^a^: Results of the Patient’s urine toxicology screening tests during treatment.

**Table 2 reports-08-00262-t002:** Urine toxicology results by key treatment periods (December 2022–March 2024).

Treatment Phase	Methadone	Cannabis	Opiates	Cocaine	Benzodiazepine	Ethanol	Interpretation
Pre-titration/ Early Treatment (First 16 months)	11/22 (−)	(+) Occasional	(+) (2×)	(−)	(+) (2×)	(−)	Absence of methadone screens is frequent despite supervised dosing, suggesting rapid clearance.
Stable Maintenance Period (December 2022– January 2024)	Consistently (−)	Occasional (+)	(−)	(+) in 5/7 tests	(−)	Detected (10 mg/dL) in some tests	Low urine creatinine levels (4 and 18 mg/dL) during the two tests points to the dilution of the urine samples; the rapid metabolism was confirmed by the serum data.
Taper/ Discontinuation Phase (February–March 2024)	(−)	(−)	(−)	(+) (5/7 tests)	(−)	Up to 10 mg/dL	

Note: ‘Ethanol level 10’ refers to a urine alcohol concentration of 10 mg/dL, measured concurrently with the urine toxicology screening.

## Data Availability

The original contributions presented in this study are included in the article. Further inquiries can be directed to the corresponding author.

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
