# Peer review of "Rapid Methadone Metabolism in Opioid Use Disorder: A Case Report of Clinical Challenges and Individualized Treatment Approaches"

_reports, 2025, doi:10.3390/reports8040262_

Round 1

Reviewer 1 Report

Comments and Suggestions for Authors

Dear Authors

I congratulate you on your case report. Here are my suggestions:

Abstract

The abstract clearly outlines the key findings. The final section regarding the patient's status after tapering to a maintenance level needs to be more precise. The text states: "After tapering doses to a maintenance level and using supervised urine collection, the patient was negative for methadone in 7 out of 7 tests (100.0%) but positive for cocaine in 5 out of 7 tests (71.4%) ." It should be clarified that this refers to the period near the end of treatment, specifically the 'Maintenance period' or a defined time frame, as the patient was also negative for methadone in 11 out of 22 toxicology screenings earlier in treatment (50.0%).

Introduction

The introduction provides a good overview of opioid use disorder, methadone maintenance therapy, and the role of CYP enzymes in metabolism.

It is noted that CYP3A4 and CYP2B6 are the major CYP isoforms involved in methadone metabolism. Later, in the "Methadone Metabolism and Serum Concentration" section, it states that CYP450 3A4 is principally in charge. While the latter is a common assertion, the slight discrepancy in emphasis should be checked for consistency or clarified as to the relative importance of each enzyme, as both are mentioned as major isoforms.

Methadone Metabolism and Serum Concentration

The information on the wide variability of methadone's half-life (8 to 59 hours) and the lack of an established therapeutic window based on serum levels is important context for the case.

The reference to the peak:trough ratios greater than 2:1 potentially pointing to rapid metabolism is a key point and should be explicitly discussed in relation to this patient's serum methadone levels (trough not detected on 4/17/19 and a peak of 170~ng/ml on 05/02/19). While the exact ratio can't be calculated without a trough on 05/02/19, the "not detected" trough on 4/17/19 strongly supports the hypothesis of rapid clearance.

Case Discussion and History of Presenting Illness

The detail provided in the patient's history is thorough. The three unintentional drug overdoses in the past, including one with methadone, are significant. This should be briefly addressed in the discussion, as it contrasts with the current presentation of rapid metabolism without OAS symptoms. Was the prior methadone overdose perhaps during the induction phase or due to drug interaction, before the pattern of rapid metabolism was established?

The fact that the patient was negative for methadone in 11 out of 22 toxicology tests during the first 16 months of treatment (50.0%) is a central finding.

The low urine creatinine levels (e.g., 11~mg/dl and 4 and 18) should be discussed as a possible indicator of a dilute sample, which could explain the negative methadone in some instances, even under supervision, though the serum test results argue against tampering.

Presentation of Tables

The manuscript contains multiple tables with overlapping or very similar information.

The first set of tables on pages 4 and 5 (un-numbered tables and Table 1) should be consolidated. Table 1 (page 5, summarizing Months 1-42 and Maintenance period) appears to be the most comprehensive. Ensure all essential data points from the unnumbered tables are included in the final, numbered tables.

Table 2 (page 5-7, Detailed Urine Toxicology Results) is very long and contains highly granular data. Given the case report format, this level of detail may not be necessary. Consider presenting a final, concise, and numbered table that summarizes the key periods (e.g., Pre-titration, Maintenance/Stable dose with negative methadone, Taper/Discontinuation with cocaine positive) and includes the relevant statistics (e.g., 11/22 negative for methadone in the first 16 months, 5/7 positive for cocaine during the maintenance period).

Ensure all final, numbered tables are appropriately cited and discussed in the text, as suggested by the peer review comments within the document.

Discussion

The discussion on potential factors—CYP2B6*5 allele polymorphism, chronic cocaine use, and demographic causes (female sex)—is a strong point and directly addresses the rapid metabolism.

The comparison to the ultra-rapid metabolizer case by Hobbins et al., who required a massive dose of 1200~mg and did show OAS symptoms, is an excellent contrast that highlights the uniqueness of this patient's stability on a relatively normal dose. This difference is the most compelling aspect of the case and should be emphasized.

Grammar/Clarity: Check for minor grammatical errors and slightly awkward phrasing (e.g., "despite adherence regular regimen" , "6 yeras ago" , "The patient may be a fast methadone metabolizer and we plan to repeat serum methadone..." ).

Consistency in Terminology: Use consistent terminology for the patient's diagnosis (e.g., Opioid Use Disorder (OUD) is used in the Abstract and Discussion , but Opioid Dependence is used in the Clinical Course section ). While related, OUD is the current DSM-5 term.

Author Response

Comment 1: It should be clarified that this refers to the period near the end of treatment, specifically the 'Maintenance period' or a defined time frame, as the patient was also negative for methadone in 11 out of 22 toxicology screenings earlier in treatment (50.0%).

Response 2: Thank you for pointing this out. We agree with this comment. Accordingly, we have clarified in the text that this refers to the period near the end of the maintenance phase. The clarification has been added on page 1, line 16.

Comment 2: The introduction provides a good overview of opioid use disorder, methadone maintenance therapy, and the role of CYP enzymes in metabolism.It is noted that CYP3A4 and CYP2B6 are the major CYP isoforms involved in methadone metabolism. Later, in the "Methadone Metabolism and Serum Concentration" section, it states that CYP450 3A4 is principally in charge. While the latter is a common assertion, the slight discrepancy in emphasis should be checked for consistency or clarified as to the relative importance of each enzyme, as both are mentioned as major isoforms.

Response 2: Thank you for highlighting this point. We have revised the text to ensure consistency regarding the relative roles of CYP3A4 and CYP2B6 in methadone metabolism. The clarification has been added on page 2, line 56.

Comment 3: Methadone Metabolism and Serum ConcentrationThe information on the wide variability of methadone's half-life (8 to 59 hours) and the lack of an established therapeutic window based on serum levels is important context for the case.The reference to the peak:trough ratios greater than 2:1 potentially pointing to rapid metabolism is a key point and should be explicitly discussed in relation to this patient's serum methadone levels (trough not detected on 4/17/19 and a peak of 170~ng/ml on 05/02/19). While the exact ratio can't be calculated without a trough on 05/02/19, the "not detected" trough on 4/17/19 strongly supports the hypothesis of rapid clearance.

Response 3: Thank you for the support. We agree with this and support the hypothesis of rapid methadone clearance.

Reviewer 2 Report

Comments and Suggestions for Authors

This is a clinically relevant and well-structured case report describing rapid methadone metabolism in a patient with opioid use disorder. The introduction is well-written and provides appropriate background supported by current literature. The discussion highlights key pharmacogenomic and behavioral mechanisms that may explain this unusual metabolic profile.

However, several aspects should be addressed to improve the clarity and scientific rigor of the manuscript:

  1. Tables and formatting: Several tables contain editorial comments and inconsistent numbering (e.g., “Months 30–36” repeated). All tables should be consistently formatted, numbered, and cited sequentially in the text. Captions should be standardized.

  2. Data presentation: While descriptive data are appropriate for a case report, a simple figure or graphical trend of methadone serum/urine levels would significantly enhance reader comprehension.

  3. Genetic analysis: The discussion mentions CYP2B6 polymorphisms as a likely factor but no genotyping was performed. Including a short justification for the lack of genetic testing would increase methodological transparency.

  4. Limitations: A dedicated limitations section is currently lacking. The authors should acknowledge the single-patient design, absence of genotyping, and potential confounding effect of concurrent cocaine use, which may complicate interpretation.

  5. Language: Although generally clear, several sentences are lengthy. Minor language editing could improve overall flow.

  6. Clinical implications: The conclusions could be expanded slightly to include practical recommendations for managing similar cases (e.g., dose adjustment or monitoring strategies for rapid metabolizers).

Addressing these points would strengthen the manuscript and make it more impactful for clinicians and researchers.

Author Response

Comment 4: Case Discussion and History of Presenting Illness

The detail provided in the patient's history is thorough. The three unintentional drug overdoses in the past, including one with methadone, are significant. This should be briefly addressed in the discussion, as it contrasts with the current presentation of rapid metabolism without OAS symptoms. Was the prior methadone overdose perhaps during the induction phase or due to drug interaction, before the pattern of rapid metabolism was established?The fact that the patient was negative for methadone in 11 out of 22 toxicology tests during the first 16 months of treatment (50.0%) is a central finding.

The low urine creatinine levels (e.g., 11~mg/dl and 4 and 18) should be discussed as a possible indicator of a dilute sample, which could explain the negative methadone in some instances, even under supervision, though the serum test results argue against tampering.

Response 4: Thank you for these valuable comments. We have incorporated a brief discussion addressing the patient’s prior methadone overdose. This have been added on page 5, lines 163–167 in the Discussion section.

Comment 5: Presentation of Tables The manuscript contains multiple tables with overlapping or very similar information.

The first set of tables on pages 4 and 5 (un-numbered tables and Table 1) should be consolidated. Table 1 (page 5, summarizing Months 1-42 and Maintenance period) appears to be the most comprehensive. Ensure all essential data points from the unnumbered tables are included in the final, numbered tables.

Table 2 (page 5-7, Detailed Urine Toxicology Results) is very long and contains highly granular data. Given the case report format, this level of detail may not be necessary. Consider presenting a final, concise, and numbered table that summarizes the key periods (e.g., Pre-titration, Maintenance/Stable dose with negative methadone, Taper/Discontinuation with cocaine positive) and includes the relevant statistics (e.g., 11/22 negative for methadone in the first 16 months, 5/7 positive for cocaine during the maintenance period).

Ensure all final, numbered tables are appropriately cited and discussed in the text, as suggested by the peer review comments within the document.

Response 5: Thank you for this helpful suggestion. We have reviewed and consolidated the tables to eliminate redundancy. The essential data from the unnumbered tables have been integrated into the final, numbered tables. Table 1 (summarizing treatment phases and urine toxicology results) has been retained on page 4, and Table 2 (detailed urine toxicology findings by key treatment periods) remains on page 5. Both tables are now appropriately cited and discussed in the revised manuscript.

Conmment: 6 Grammar/Clarity: Check for minor grammatical errors and slightly awkward phrasing (e.g., "despite adherence regular regimen" , "6 yeras ago" , "The patient may be a fast methadone metabolizer and we plan to repeat serum methadone..." ).

Response 6: Thank you for pointing out these grammatical and clarity issues. The phrase “despite adherence regular regimen” has been corrected to “despite adherence to a regular regimen” on page 1, line 11. The sentence regarding methadone metabolism has been revised for clarity on pages 3–4, lines 132–134, and the spelling of “years” has been corrected throughout the manuscript.

Comment 6: Consistency in Terminology: Use consistent terminology for the patient's diagnosis (e.g., Opioid Use Disorder (OUD) is used in the Abstract and Discussion , but Opioid Dependence is used in the Clinical Course section ). While related, OUD is the current DSM-5 term.

Response 7: Thank you for noting this inconsistency. The term “Opioid Dependence” has been replaced with “Opioid Use Disorder (OUD)” to ensure consistency with DSM-5 terminology. This correction has been made on page 3, line 112.

Reviewer 3 Report

Comments and Suggestions for Authors

This case report presented an important case report in the Addiction Biology which may solve some unsolved pieces of puzzles in Methadone Mechanism of Action. In fact, the right fit of this case report should be in Pharmacogenomics and Personalized Medicine which is mentioned partially in its title "...  Individualized Treatment Approaches". Genetically, the case background clearly indicates that she has a various genetic associations with multiple phenotypes and thus, to present such this case, the authors should have utilized at least a targeted genotyping on her Pharmacogenes. 
My overall standpoint is that without a genetic testing, this interesting case may be neglected by future studies. Therefor, I highly recommend this paper for a major revision with additional molecular evidences such as Whole-Exome Sequencing, Targeted Pharmacogenomics panel (genotyping important CYP family genetic variants), or at least Genotyping of "CYP2B6*5 polymorphism" according to the author's claims in the Discussion section. 
Finally, as a minor comment, please check this out: 

  • In lines 39-41, the statement concerning CYP2D6 “with close to no impact on methadone dosage requirements” is not align with Pharmacogenomics data. Based on ClinPGx (formerly PharmGKB database), CYP2D6 has 6 significant PGx annotations with Methadone. Please use the link https://www.clinpgx.org/gene/PA128/variantAnnotation and rewrite the aforementioned part corrected.

Author Response

Comment 4 Limitations: A dedicated limitations section is currently lacking. The authors should acknowledge the single-patient design, absence of genotyping, and potential confounding effect of concurrent cocaine use, which may complicate interpretation.

Response 4: Limitations

We agree with this comment. A dedicated limitations section has been included, acknowledging the single-patient design, absence of genotyping, and potential confounding from concurrent cocaine use.Added on page 7, lines 194–199.

Comment 5 Language: Although generally clear, several sentences are lengthy. Minor language editing could improve overall flow.

Response 5: Language

Thank you for the observation. The manuscript has undergone thorough language revision to improve clarity and sentence flow, particularly by shortening lengthy sentences and refining transitions. Edits made throughout the manuscript.

Comment 6: Clinical implications: The conclusions could be expanded slightly to include practical recommendations for managing similar cases (e.g., dose adjustment or monitoring strategies for rapid metabolizers).

Response 6: Clinical implications

We appreciate this insightful suggestion. The conclusion has been expanded to include practical recommendations for managing rapid methadone metabolizers, such as individualized dose adjustment, close serum level monitoring, and consideration of pharmacogenetic testing in similar cases. Added on page 7, lines 212–215.

Round 2

Reviewer 2 Report

Comments and Suggestions for Authors

Dear Authors,

Thank you for submitting your revised manuscript entitled “Rapid Methadone Metabolism in Opioid Use Disorder: A Case Report of Clinical Challenges and Individualized Treatment Approaches.”

Your revisions have substantially improved the paper’s structure, clarity, and interpretative depth. The inclusion of additional data tables, expanded discussion on pharmacogenetic aspects, and clear acknowledgment of study limitations represent significant progress.

However, several minor issues remain before the paper can be considered ready for publication:

  1. Language and formatting

    • Please carefully proofread the manuscript. Numerous duplicated terms remain (e.g., “patientPatient,” “methadoneMethadone”), likely due to tracked-change artifacts.

    • Some tables (especially Table 1 and Table 2) require formatting corrections — unify headers, font size, and alignment.

    • Verify that all tables are cited in numerical order in the text.

  2. Interpretation

    • Certain sentences overstate causality between cocaine use and methadone metabolism (e.g., “confirmed rapid metabolism”). These should be moderated to “suggestive of” or “consistent with.”

    • The discussion should emphasize that the observed findings are clinically compatible with rapid metabolism but not genetically confirmed.

  3. Methodological clarity

    • Please add concise information on how urine and serum methadone levels were measured (e.g., assay type, detection threshold, and timing).

    • Ensure that all laboratory values use consistent units (e.g., ng/mL).

  4. Language quality

    • While the English is understandable, a professional language editing service is recommended to correct residual grammatical errors and improve readability.

Once these minor adjustments are made, your paper will meet the standards of Reports and provide valuable insight into individualized methadone treatment and pharmacokinetic variability.

Author Response

Comment 1 Tables and formatting: Several tables contain editorial comments and inconsistent numbering (e.g., “Months 30–36” repeated). All tables should be consistently formatted, numbered, and cited sequentially in the text. Captions should be standardized.

Response 1: Tables and formatting

Thank you for this valuable observation. All tables have been reviewed for consistency. Repetitive labels such as “Months 30–36” have been corrected, and tables are now sequentially numbered and cited appropriately in the text. Captions have been standardized across all tables.
Changes made on page 4–5 as table 1 and table 2

Comment 2 Data presentation: While descriptive data are appropriate for a case report, a simple figure or graphical trend of methadone serum/urine levels would significantly enhance reader comprehension.

Response 2: Data presentation

We appreciate this helpful suggestion. A simple tablse illustrating the trend of serum and urine methadone levels has been added to enhance clarity and visualization of the patient’s pharmacokinetic profile Changes made on page 4–5 as table 1 and table 2

Comment 3 Genetic analysis: The discussion mentions CYP2B6 polymorphisms as a likely factor but no genotyping was performed. Including a short justification for the lack of genetic testing would increase methodological transparency.

Response 3: Genetic analysis

Thank you for the suggestion. A short justification for the absence of genotyping has been added to improve methodological transparency, Added on page 6, lines 187-189.

Reviewer 3 Report

Comments and Suggestions for Authors

The authors addressed my concerns and I suggest this version for acceptance. Good luck.

Author Response

This case report presented an important case report in the Addiction Biology which may solve some unsolved pieces of puzzles in Methadone Mechanism of Action. In fact, the right fit of this case report should be in Pharmacogenomics and Personalized Medicine which is mentioned partially in its title "...  Individualized Treatment Approaches". Genetically, the case background clearly indicates that she has a various genetic associations with multiple phenotypes and thus, to present such this case, the authors should have utilized at least a targeted genotyping on her Pharmacogenes. 
My overall standpoint is that without a genetic testing, this interesting case may be neglected by future studies. Therefor, I highly recommend this paper for a major revision with additional molecular evidences such as Whole-Exome Sequencing, Targeted Pharmacogenomics panel (genotyping important CYP family genetic variants), or at least Genotyping of "CYP2B6*5 polymorphism" according to the author's claims in the Discussion section. 
Finally, as a minor comment, please check this out: 

Comment 1: In lines 39-41, the statement concerning CYP2D6 “with close to no impact on methadone dosage requirements” is not align with Pharmacogenomics data. Based on ClinPGx (formerly PharmGKB database), CYP2D6 has 6 significant PGx annotations with Methadone. Please use the link https://www.clinpgx.org/gene/PA128/variantAnnotation and rewrite the aforementioned part corrected.

Response 1: Correction incorporated on page 2, line 40, to align with ClinPGx pharmacogenomic data regarding CYP2D6 and its role in methadone metabolism.